# Role of Diffusion Tensor Imaging in Diagnosis and Estimation of Shunt Effect for Hydrocephalus in Stroke Patients: A Narrative Review

**DOI:** 10.3390/diagnostics12061314

**Published:** 2022-05-25

**Authors:** Sung-Ho Jang, Min-Jye Cho

**Affiliations:** Department of Physical Medicine and Rehabilitation, College of Medicine, Yeungnam University, Namku, Taegu 42415, Korea; strokerehab@hanmail.net

**Keywords:** hydrocephalus, diffusion tensor imaging, stroke, shunt, intracerebral hemorrhage

## Abstract

Hydrocephalus is a dilatation of the brain ventricular system by the accumulation of cerebrospinal fluid within the ventricle caused by impaired cerebrospinal fluid circulation or clearance. A diagnosis of hydrocephalus at the chronic stage of stroke has been mainly made by clinical features and radiologic findings on brain computed tomography and magnetic resonance imaging. On the other hand, it could not determine the effect of hydrocephalus or shunt effect on the periventricular neural structures. By contrast, these effects on the periventricular neural structures can be estimated using diffusion tensor imaging (DTI). This article reviewed 10 DTI-based studies related to the diagnosis and estimation of the shunt effect for hydrocephalus in stroke patients. These studies suggest that DTI could be a useful diagnostic and estimation tool of the shunt effect for hydrocephalus in stroke patients. In particular, some studies suggested that fractional anisotropy value in the periventricular white matter could be a diagnostic biomarker for hydrocephalus. As a result, the role of DTI in diagnosing and estimating the shunt effect for hydrocephalus in stroke patients appears to be promising. However, the number of studies and patients of all reviewed studies were limited (10 studies including a total of 58 stroke patients with heterogenous brain pathologies).

## 1. Introduction

Hydrocephalus is a dilatation of the brain ventricular system caused by the accumulation of cerebrospinal fluid within the ventricle from impaired cerebrospinal fluid circulation or clearance [1,2,3,4,5]. This excessive cerebrospinal fluid increases pressure on the periventricular neural structures. Consequently, acute hydrocephalus could cause various clinical symptoms including headaches, vomiting, nausea, sleepiness, coma, papilledema, and diplopia [6]. By contrast, chronic or normal pressure hydrocephalus is accompanied by various clinical features, including the clinical triad (gait disturbance, cognitive impairment, and urinary incontinence) [1,2,3,4,5,7,8]. The diagnosis of hydrocephalus at the chronic stage of stroke has generally been made mainly by clinical features and radiologic findings (computed tomography (CT) and conventional magnetic resonance imaging (MRI)) [1,2,3,5,7,9,10]. Although measurements of the relative ventricular size have been used as an objective evaluation tool for the radiology findings, it could not precisely determine the effect of hydrocephalus on the periventricular neural structures [9,11]. Furthermore, the effects of surgical interventions for hydrocephalus on the periventricular neural structures also could not be determined precisely. 

Diffusion tensor imaging (DTI), which generates images based on estimations of the diffusion characteristics of water molecules in the brain microstructures, has enabled an evaluation of the microstructural features of the white matter [12,13]. As a result, DTI can estimate the effects of the hydrocephalus or surgical interventions for hydrocephalus on the periventricular white matter or neural structures using DTI parameters [5,8,14,15]. DTI parameters are measured using region of interest (ROI) methods for specific neural areas or an analysis of the whole subcortical white matter using specific analysis programs, such as tract-based spatial statistics [12,13,16]. By contrast, the neural tracts can be reconstructed three-dimensionally using diffusion tensor tractography (DTT) reconstructed based on DTI data [17,18,19,20]. The main advantage of DTT is that the entire neural tract can be evaluated from DTT parameters and configurational analysis (integrity and configuration) [17,18,20]. Consequently, the effects of the hydrocephalus or surgical interventions for hydrocephalus on the periventricular neural tracts or structures can be estimated using DTT [21,22,23,24,25,26]. Fractional anisotropy (FA: the state of white matter organization because it is a measure of the degree of directionality and integrity of the white matter microstructures), apparent diffusion coefficient (or mean diffusivity)(the magnitude of water diffusion), and tract volume (or fiber number; the number of voxels within a neural tract which represents the number of fibers within of a neural tract) have been commonly used as DTI or DTT parameters [12,13,17,18,20]. As a result, DTI or DTT for periventricular white matter or neural structures have been suggested as a non-invasive tool for evaluating the presence, severity, and shunt effect of hydrocephalus in stroke patients [4,8,14,15,21,22,23,24,25,26].

This study reviewed 10 DTI or DTT-based studies that demonstrated the usefulness of DTI or DTT for evaluating the presence, severity, and shunt effect of hydrocephalus in stroke patients (Table 1) [4,8,14,15,21,22,23,24,25,26]. This review aims to present the role of DTI in the diagnosis and estimation of the shunt effect for hydrocephalus in stroke patients.

## 2. Methods

DTI or DTT-based studies that investigated the diagnosis and estimation of shunt effect for hydrocephalus in stroke patients were searched. The electronic databases (Google Scholar and MEDLINE database (PubMed)) were used to search the relevant studies published from 1992 until 31 March 2022. The search strategy for identifying potentially relevant articles was based on the subject heading and keywords/abbreviations with synonyms (DTI, DTT, diagnosis, estimation, shunt, effect, hydrocephalus, stroke, hemorrhage, and infarction). The inclusion criteria for subjects were limited to studies involving human subjects with stroke. Overall, 10 studies were selected for review [4,8,14,15,21,22,23,24,25,26].

## 3. Role of Diffusion Tensor Imaging in the Diagnosis of Hydrocephalus in Stroke Patients

In 2010, Osuka et al. examined whether ventromegaly is related to true hydrocephalus or brain atrophy in patients who exhibited ventromegaly after brain injury [8]. Ten patients with chronic hydrocephalus (eight with subarachnoid hemorrhage (SAH) and two with idiopathic normal pressure hydrocephalus) and eight patients with brain atrophy were recruited. All patients with chronic hydrocephalus showed improvement of the clinical features of hydrocephalus after ventriculoperitoneal or lumboperitoneal shunt surgery. The FA values were measured in 10 ROIs in the periventricular areas: the corpus callosum (the splenium, anterior third of the body, posterior third of the body, and genu), caudate nucleus, thalamus, anterior and posterior limbs of the internal capsule, the periventricular corona radiata, and high-intensity area in the periventricular frontal white matter. Only the FA value of the caudate nucleus was higher in the hydrocephalus group than in both the atrophy and normal control groups. The authors attributed the increased FA value in the caudate nucleus to tissue compression by the hydrocephalus [4,14,15,19]. The FA value of the caudate nucleus decreased in all patients in the hydrocephalus group cases after shunt surgery. By contrast, inconsistent results were observed in the other ROIs. As a result, the authors concluded that the FA value of the caudate nucleus could be a diagnostic biomarker for differentiating true hydrocephalus from brain atrophy. To the best of the authors’ knowledge, this is the first study to demonstrate the role of DTI in the diagnosis of hydrocephalus in stroke patients. On the other hand, the heterogeneous brain pathologies of the recruited patients were a limitation of this study. This study was classified as the diagnosis part of this review because the authors focused on the diagnostic utility of DTI, even though this study included DTI data after shunt surgery [8]. 

In 2013, Jang et al. investigated the effects of hydrocephalus on the neural structures in the periventricular area in 14 patients diagnosed with hydrocephalus after a spontaneous intracerebral hemorrhage (ICH) [15]. DTI was scanned at the chronic stage of ICH (5–52 weeks after onset) and DTI parameters were estimated in the six ROIs: the anterior corona radiata, posterior corona radiata, genu of the corpus callosum, splenium of the corpus callosum, anterior limb of the internal capsule, and posterior limb of the internal capsule. The FA value increased only in the anterior corona radiata without changes in the other ROIs and apparent diffusion coefficient values. The authors assumed that an increase in FA value in the anterior corona radiata indicated greater fiber packing caused by mechanical pressure in this area by the hydrocephalus [8,15,19]. On the other hand, the authors measured the relative width ratios between the maximum distance of the ventricular walls and the maximum width of the brain at the anterior horn, ventricular body, and posterior horn of the lateral ventricle [9,11]. The patient group showed higher relative width ratios in three areas than the control group. In particular, the relative width ratios of the anterior horn were increased more than the other relative width ratios. As a result, the authors concluded that in patients with hydrocephalus following ICH, the anterior corona radiata was compressed more by hydrocephalus than the other neural structures of the periventricular area [4,8,12,13,14,15]. They suggested that the FA value of the anterior periventricular corona radiata could provide an important diagnostic clue for hydrocephalus. The advantage of this study was that the authors suggested a pathophysiological mechanism for the clinical features of hydrocephalus related to a frontal lobe dysfunction, such as gait disturbance, cognitive impairment, and urinary incontinence [27,28,29]. Nevertheless, this study was limited by the small number of subjects.

In 2016, Jang and Lee examined the utility of the distance between corticospinal tracts on DTT as a diagnostic biomarker for hydrocephalus in stroke patients [21]. Fifteen patients who underwent shunt surgery for hydrocephalus that developed after spontaneous intraventricular hemorrhage (IVH) were recruited. DTI scanning was performed at the chronic stage (more than one month after onset and before shunt surgery). The authors measured two types of distances on the axial slice of the corona radiata level, which was the widest distance between the corticospinal tracts: the absolute distance (between the most medial point of both corticospinal tracts in the mediolateral horizontal direction) and the relative distance (absolute distance divided by the distance between both lateral margins of the brain at the same horizontal line of both corticospinal tracts on the same axial image). Both absolute and relative distances or the corticospinal tracts were higher in the patient group than in the normal control group. The authors suggested that the absolute and relative distances of the corticospinal tracts on DTT could be a diagnostic biomarker of hydrocephalus because the corticospinal tract descended bilaterally through the corona radiata which could easily be affected by hydrocephalus [30]. Therefore, the advantage of this study was that the authors suggested that the distance of the neural tracts in the periventricular white matter could be a diagnostic biomarker for hydrocephalus and ventricular size. On the other hand, the authors did not report the real effect of hydrocephalus on the corticospinal tracts using DTI parameters. Furthermore, they did not compare the diagnostic value with the ventricular size currently used as a diagnostic biomarker for hydrocephalus [9,11]. 

## 4. Role of Diffusion Tensor Imaging in Estimating the Shunt Effect for Hydrocephalus in Stroke Patients 

In 2011, Jang and Kim reported a stroke patient with hydrocephalus who exhibited changes in the neural structures in the periventricular area on DTI after shunt surgery [14]. A 48-year-old male underwent aneurysm clipping and hematoma removal after rupturing the middle cerebral artery bifurcation aneurysm. Before shunt surgery at three months after onset, he presented gait inability and cognitive impairment (Mini-Mental State Exam (MMSE): eight points, cut-off score < 25, and full score 30) and urinary incontinence [31]. The patient underwent ventriculoperitoneal shunt surgery for hydrocephalus. After shunt surgery, his cognition was improved greatly to 24 points on MMSE with the disappearance of urinary incontinence, and he could walk independently after eight weeks of rehabilitation following the shunt surgery. DTI was performed three times in the patient (first DTI: before the shunt, three months after onset; second DTI: two weeks after shunt; third DTI: eight weeks after shunt). The FA values in the left corona radiata, corticospinal tract, and arcuate fasciculus were increased by 16%, 5%, and 4%, respectively, than the normal control subjects, and these FA values were decreased by 26%, 17%, and 4%, respectively, on post shunt-two week DTT. The FA values at two and eight weeks post-shunt were similar. By contrast, the apparent diffusion coefficient values of the three DTIs in all three neural structures were similar to those of the normal control subjects. As a result, the authors concluded that the closer the neural structures to the lateral ventricle showed larger increases and greater decreases of the FA values before and after the shunt surgery, respectively, indicating that the closer neural structures to the lateral ventricle were affected more by the pressure of the lateral ventricle than structures further away. The authors suggested that DTI parameters would be a useful biomarker for diagnosing and determining the effect of shunt surgery for hydrocephalus. The advantage of this study was that the authors demonstrated that the FA value in the adjacent neural structures to the lateral ventricle area was the most sensitive biomarker for diagnosis and evaluation of shunt effect for hydrocephalus. On the other hand, this study is limited because it is a case report. 

In 2012, Scheel et al. investigated changes of DTI parameters according to shunt surgery in patients with hydrocephalus [4]. Thirteen patients with chronic hydrocephalus were recruited (seven patients with idiopathic normal pressure hydrocephalus, three patients with SAH, and three patients with aqueduct stenosis). DTI was performed twice before (1–35 days) and after (9–42 weeks) ventriculoperitoneal shunt surgery. DTI was analyzed by both ROI method and whole subcortical white matter analysis using Tract-Based Spatial Statistics [16]. The DTI parameters were measured in nine ROIs: the corticospinal tract (the subcortical white matter of the precentral gyrus, periventricular white matter, posterior limb of the internal capsule, midbrain, and lower pons), caudate nucleus, and corpus callosum (genu, body, and splenium). In the patient group before shunt surgery, the FA values and parallel diffusivity (a kind of diffusivity) were increased in the corticospinal tract, particularly at the posterior limb of the internal capsule. In contrast, the FA value decreased in the corpus callosum with increasing parallel and radial diffusivities compared to the normal control group. After shunt surgery, all DTI parameters showed a trend towards normalization although there were no significant differences between the patient and normal control groups. The authors suggested that the increase in FA value in the posterior limb of the internal capsule appeared to be very specific for hydrocephalus. As a result, the authors concluded that the changes in DTI parameters were dependent on the brain regions and required careful interpretation of DTI parameters as for diagnostic or follow-up assessments in patients with hydrocephalus. The authors identified the specific DTI parameters and brain area for the diagnosis and follow-up for hydrocephalus using both the ROI and the whole subcortical white matter analysis methods. On the other hand, this study was limited by the small number of subjects with heterogeneous brain pathologies and DTI scanning times (post-shunt DTI scanning: 9–42 weeks after shunt surgery). 

In 2016, Lee and Jang reported a stroke patient who presented changes in the cinguli along with an improvement of the impaired cognition after shunt surgery for hydrocephalus [22]. A 59-year-old female was diagnosed with ICH and IVH after a rupture of an arteriovenous malformation in the thalamus. When she started rehabilitation four weeks after onset, she showed severely impaired cognition as four points on MMSE and brain MRI revealed hydrocephalus [31]. Although she underwent comprehensive rehabilitation for two weeks, her impaired cognition did not improve due to hydrocephalus. As a result, she underwent ventriculoperitoneal shunt surgery. Her severely impaired cognition revealed significant improvement of 24 points on MMSE at seven days and 27 points on MMSE at four weeks after shunt surgery [31]. On pre-shunt DTT, discontinuations were observed between both anterior cinguli and both basal forebrains. On the other hand, the discontinued anterior portions of both cinguli were elongated anteriorly on the post-shunt DTT. In particular, the discontinued left cingulum was connected to the left basal forebrain, which contains the cholinergic nuclei [32,33,34]. The authors assumed that restoring the disconnection between the left anterior cingulum and the left basal forebrain was compatible with improving impaired cognition after shunt surgery [34]. As a result, the authors attributed the reconnection of the discontinued left anterior cingulum with the basal forebrain to decompression of hydrocephalus pressure on the periventricular white matter in this patient. The authors suggested that DTT could help estimate the neural tract state and effect after shunt surgery for hydrocephalus. The advantage of this study was that the authors demonstrated changes in the neural tracts following shunt surgery with changes to the corresponding clinical feature. Nevertheless, this study was a case report, and DTT parameter data were not provided. 

In 2017, Jang et al. reported a stroke patient who presented with improved akinetic mutism and restoration of the prefronto-caudate tract after shunt surgery [23]. A 76-year-old female was diagnosed with a SAH following the rupture of a posterior communicating artery aneurysm. She showed the typical clinical features of akinetic mutism (no spontaneous movement or speaking) six months after onset, and brain MRI revealed hydrocephalus [35,36]. Although she underwent comprehensive rehabilitation for two months, she did not show any improvement of the akinetic mutism. She underwent ventriculoperitoneal shunt surgery for hydrocephalus eight months after onset. After one month of rehabilitation from shunt surgery, her akinetic mutism showed some improvement: she could perform some daily activities herself and could speak with some fluency. On pre-shunt DTT (two months before shunt surgery), the prefronto-caudate tract revealed discontinuations in both hemispheres, whereas the integrities of the discontinued prefronto-caudate tracts were restored on post-shunt DTT (one month after shunt surgery and nine months after onset). The integrities of both arcuate fasciculi were well preserved on the pre- and post-shunt DTTs. As a result, the authors concluded that the relief of hydrocephalus by shunt surgery was the primary reason for the restored compressed prefronto-caudate tracts by the hydrocephalus. The authors demonstrated that her mutism was related to the prefronto-caudate tract by demonstrating both intact arcuate fasciculi on both DTTs. On the other hand, DTIs were scanned slightly far from shunt surgery, and no DTT parameter data was provided. 

Jang et al. (2018) reported a stroke patient who showed restoration of the corticoreticular pathways and regained gait function after shunt surgery for hydrocephalus [24]. A 67-year-old female underwent stereotactic drainage for IVH following a rupture of the posterior communicating artery. When she started rehabilitation four weeks after onset, hydrocephalus was detected on the brain CT (Figure 1). She presented with quadriparesis (Medical Research Council, both arms: 3–3^+^, both legs: 2–2^+^) and could not even stand [37]. Although she underwent comprehensive rehabilitation for three weeks, quadriparesis was not recovered, and she underwent ventriculoperitoneal shunt surgery. Her quadriparesis revealed recovery five days after shunt surgery (Medical Research Council, both arms: 4, both legs: 3–4) [37]. As a result, she could walk with mild assistance on an even floor at five days and walk on stairs at four weeks after shunt surgery. On pre-shunt DTT (three weeks before shunt surgery), both corticoreticular pathways showed discontinuations (the right corticoreticular pathway at the subcortical white matter level, and the left corticoreticular pathway at the midbrain level) [38]. By contrast, on post-shunt DTT (five days after shunt surgery), the discontinued integrities of the corticoreticular pathway were restored to the premotor cortex in both hemispheres (Figure 1). As a result, the authors concluded that her gait recovery, which was responsible for the corticoreticular pathway function, appeared to be consistent with the change in corticoreticular pathway on DTTs following shunt surgery [39,40]. Furthermore, the authors attributed the restoration of the discontinued corticoreticular pathways on the post-shunt DTT to decompression of the pressure on the corticoreticular pathways, which were severely compressed by the high intraventricular pressure. The demonstration of gait recovery with changes in the corticoreticular pathway after shunt surgery, despite no recovery during rehabilitation before shunt surgery, was an advantage of this study. Nevertheless, the authors provided only configurational changes in the corticoreticular pathway without DTT parameters in a case report. 

In 2019, Jang and Lee reported a stroke patient who showed changes in the ascending reticular activating system concurrent with the recovery of impaired consciousness after shunt surgery for hydrocephalus [25]. A 65-year-old female was diagnosed with a SAH due to rupture of the posterior communicating artery and underwent extraventricular drainage for IVH. After six months after onset, when she started rehabilitation, she presented with severely impaired consciousness, with a Glasgow Coma Scale (GCS) score of 7 [41]. Hydrocephalus was observed on the brain MRI. She underwent comprehensive rehabilitation, and her GCS score had recovered to 9 at 20 days after starting the rehabilitation [41]. The patient underwent ventriculoperitoneal shunt surgery for the hydrocephalus. After shunt surgery, her consciousness was improved as GCS 11 at 1 day and 15 (full score) at 10 days after shunt surgery. DTI data were acquired twice (6 and 7 months after onset). The neural connectivity of the upper ascending reticular activating system from the thalamic intralaminar nuclei to the prefrontal cortex was increased in both hemispheres on the post-shunt DTT (approximately 10 days after shunt surgery) compared to pre-shunt DTT (approximately 20 days before shunt surgery) [42,43]. The increased connectivity to both prefrontal cortices on post-shunt DTT, which are important brain areas for consciousness, coincided with the recovery of impaired consciousness after shunt surgery [44,45]. As a result, the increased neural connectivity to the prefrontal cortex was attributed to decompression of the pressure on the periventricular white matter in this patient. The authors suggest the necessity of DTT for evaluating the ascending reticular activating system in patients with disorders of consciousness caused by hydrocephalus and for evaluating the effects of the shunt surgery for hydrocephalus. The advantage of this study was that the importance of evaluating the ascending reticular activating system for patients with disorders of consciousness and hydrocephalus was demonstrated. Nevertheless, this study was pre-shunt DTT taken a little far from shunt surgery (20 days before shunt surgery) and this was a case report.

Recently, Cho and Jang (2022) reported a stroke patient who presented with improved cognitive impairment with changes in the Papez circuit after shunt surgery for hydrocephalus [26]. A 50-year-old male was diagnosed with spontaneous ICH, SAH, and IVH. He underwent coiling for a ruptured anterior communicating artery aneurysm, stereotactic drainage, and decompressive craniectomy. At three months after onset, his cognition was severely impaired as six points on MMSE (two days before shunt surgery) [31]. He was diagnosed with hydrocephalus and underwent ventriculoperitoneal shunt surgery. After shunt surgery, his cognitive impairment improved significantly to 17 points on MMSE (six days after shunt surgery) [31]. On pre-shunt DTT (two days before shunt surgery), the neural tracts of the Papez tract, except for the left mammillothalamic tract, were not reconstructed [46]. On the other hand, the left mamillothalamic tract was thickened, and the left thalamocingulate tract, fornix, and both cinguli were reconstructed on the post-shunt DTT (6 days after shunt surgery). In addition, the FA value and tract volume of the left mamillothalamic tract was increased on post-shunt DTT compared to pre-shunt DTT. As a result, the reconstruction of the thalamocingulate tract, fornix, cingulum, and mamillothalamic tract on post-shunt DTT, which were not reconstructed on pre-shunt DTT, indicated restoration of these compressed neural tracts after the relief of high intraventricular pressure caused by hydrocephalus via the shunt surgery. In addition, the increased FA and tract volume value of the left mamillothalamic tract on post-shunt surgery indicated increased directionality and a larger number of neural fibers in these neural tracts, respectively [12,13]. The DTT-observed restoration of the Papez circuit coincided with the significant improvement in the patient’s MMSE score. The authors demonstrated that successful shunt surgery improved the Papez circuit state in a patient with cognitive impairment caused by hydrocephalus. In addition, the changes in DTT parameters with configurational changes in the Papez circuit were reported. Nevertheless, this study was a case report.

## 5. Conclusions

This article reviewed 10 DTI-based studies related to the diagnosis (three studies) and estimation of the shunt effect (seven studies) for hydrocephalus in stroke patients [4,8,14,15,21,22,23,24,25,26]. These studies suggest that DTI could be a useful diagnostic tool for hydrocephalus in stroke patients. In particular, some studies suggested that the FA value in the periventricular white matter, particularly the neural structures around the anterior horn, could be a diagnostic biomarker for hydrocephalus [4,8,14,15]. Follow up DTTs for the neural tracts related to the clinical features of hydrocephalus could be a useful estimation tool of the shunt effect for hydrocephalus in stroke patients [4,14,22,23,24,25,26]. As a result, the role of DTI in diagnosis and estimation of the shunt effect for hydrocephalus in stroke patients appears to be promising. On the other hand, the 10 DTI-based studies for hydrocephalus in stroke patients are much fewer than those on hydrocephalus in other brain pathologies, including idiopathic normal pressure hydrocephalus [4,8,14,15,21,22,23,24,25,26,47,48]. In addition, six of the ten studies were case reports. Therefore, further prospective studies recruiting a large number of patients will be needed. 

This review has some limitations. First, although the reviewed studies were included through the search strategy, the number of studies and patients of all reviewed studies were limited (10 studies including a total of 58 stroke patients). Second, the patients with heterogenous brain pathologies such as SAH, ICH, and IVH because stroke comprises these brain pathologies. Hence, further prospective studies involving homogenous brain pathologies should be encouraged.

## Figures and Tables

**Figure 1 diagnostics-12-01314-f001:**
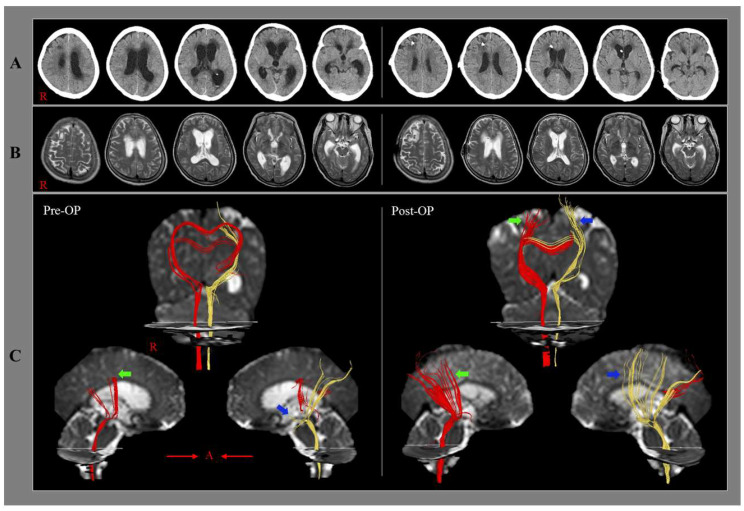
Changes of the corticoreticular pathway following shunt operation. Pre-op brain CT images (**A**) and MR images (**B**) show dilatation of the ventricular system and decrement of dilatation of the ventricular system after shunt operation. Results of diffusion tensor tractography (DTT) for the CRP (**C**). On the pre-op DTT, discontinuations (the right CRP: at subcortical white matter level-green arrow, and the left CRP: at the midbrain level-blue arrow) of the CRP fibers from the premotor cortex were observed in both hemispheres. On the post-op DTT, the CRP fibers were elongated to the respective premotor cortex in both hemispheres (arrows) (reprinted with permission from Medicine (Baltimore):2018;97(4):e9512).

**Table 1 diagnostics-12-01314-t001:** Diffusion tensor imaging studies related to diagnosis and estimation of shunt effect for hydrocephalus in stroke patients.

Classification	Publication	Patient No.	Stroke Pathology	Analyzed Neural Structures	Results
Diagnosis	Osuka et al. (2010) [8]	10	8 patients (SAH)	Caudate nucleus Corpus callosum Thalamus Internal capsule CR Frontal white matter	FA ↑ (Caudate nucleus)
	Jang et al. (2013) [15]	14	ICH	Anterior and posterior CR Corpus callosum Imternal capusle	FA ↑ (Anterior CR)
	Jang and Lee (2016) [21]	15	IVH	CST	Distance ↑ (CST)
Effect of shunt surgery	Jang and Kim (2011) [14]	1	ICH	CR CST Arcuate fasciculus	CR (FA 16% ↑ → 26% ↓) CST (FA 5% ↑ → 17% ↓) Arcuate fasciculus (FA 4% ↑ → 4% ↓)
	Scheel (2012) [4]	13	3 patients (SAH)	CST Caudate nucleus Corpus callosum	CST (FA ↑ → Normalization) Corpus callosum (FA ↓ → Normalization)
	Lee and Jang (2016) [22]	1	ICH IVH	Cingulum	Integrity restoration of cingulum
	Jang et al. (2017) [23]	1	SAH	Prefronto-caudate tract	Integrity restoration of prefronto-caudate tract
	Jang et al. (2018) [24]	1	IVH	Corticoreticular pathway	Integrity restoration of corticoreticular pathway
	Jang and Lee (2019) [25]	1	SAH	Ascending reticular activating system	Neural connectivity to prefrontal cortex ↑
	Cho and Jang (2022) [26]	1	ICH IVH SAH	Papez circuit	Integrity restoration of papez circuit

SAH: subarachnoid hemorrhage, CR: corona radiata, FA: fractional anisotropy, ICH: intracerebral hemorrhage, IVH: intraventricular hemorrhage, CST: corticospinal tract.

## Data Availability

Not applicable.

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
