# Peer review of "Role of Diffusion Tensor Imaging in Diagnosis and Estimation of Shunt Effect for Hydrocephalus in Stroke Patients: A Narrative Review"

_diagnostics, 2022, doi:10.3390/diagnostics12061314_

Round 1
Reviewer 1 Report
This manuscript reviews 10 papers including case reports and very small cohorts, number of them obviously published by the first author. This is more like a listing of the main finding of the previous case reports than a synthesis producing new information. After “this listing” the paper goes directly to conclusions without appropriate general discussion. At least the methodological limitations (beyond the very limited number of these remarkably heterogenious cases) should be addressed.
Minor notes:
Introduction, lines 36-37: “…hydrocephalus is accompanied by various clinical features, including the clinical triad (gait disturbance, cognitive impairment, and urinary incontinence) [1-7]” These symptoms indicate specifically “normal pressure hydrocephalus” or chronic hydrocephalus. Please be specific and state NPH (or chronic HC) or indicate also the typical symptoms of acute hydrocephalus as well.
Lines 39-40: ultrasonography is useless in adults
Table 1 layout should be corrected, in the current form the columns are messy and thus difficult to follow.
Lines 188-189: “…but there were no significant differences to the normal control group remained.” Please rewrite.
Reviewer 2 Report
The authors analyzed 10 studies with a total of 58 patients on the use of DTI and tractography in patients with hydrocephalus. Unfortunately, only very small and limited case series have been published on this topic. Moreover, the inclusion criteria were very heterogeneous (SAH, ICH, ...), which severely limits the usefulness for clinical routine. Nevertheless, the review provides a very good overview of the current state of research, on which prospective studies could now be conducted. The methods of the literature review should be outlined. Some aspects of the manuscript could be improved:
- Some parts of the introduction are too unspecific, they should be corrected. In line 35-38, the authors stated that hydrocephalus is associated with increased intracranial pressure. In the very next sentence, they declared the signs of the normal pressure hydrocephalus as the clinical characteristics - but this does not fit to the content before.
- It is not correct that diagnosis of hydrocephalus in made by ultrasound in routine practice (line 39).
- In Table 1, the authors presented the analyzed studies (n=10) including 58 patients in total. Thus, only very small and limited case series have been published related to this topic. Additionally, the inclusion criteria were very heterogenous (SAH, ICH, …)
- The pathophysiological associations described by Jang an Lee et al. 2016 (line 125-143) are not clearly presented. It is not clear for the reader why distances (?) of the CST can be used as predictors for hydrocephalus.
- The following sentence is confusing and should be clarified: „the neural structures closer to the lateral ventricle showed larger increases and greater decreases of the FA values before and after the shunt surgery, respectively,“ (line 164-165).
- The summary is a bit verbose and imprecise. The main finding that there are only very small studies so far, which do not allow a recommendation for routine work, but are a good basis for prospective studies, should be stated more concisely.
Round 2
Reviewer 1 Report
My concerns have been addressed. I just ask to add the time range of the litterature search (from XXX until March 31 2022).
Author Response
Point 1) My concerns have been addressed. I just ask to add the time range of the literature search (from XXX until March 31 2022).
Answer: Thank you for the reviewer’s comment. We revised as follows.
- Methods
DTI or DTT-based studies that investigated the diagnosis and estimation of shunt effect for hydrocephalus in stroke patients were searched. The electronic databases [Google Scholar and MEDLINE database (PubMed)] were used to search the relevant studies published from 1992 until March 31, 2022. The search strategy for identifying potentially relevant articles was based on the subject heading and keywords/abbreviations with synonyms (DTI, DTT, diagnosis, estimation, shunt, effect, hydrocephalus, stroke, hemorrhage, and infarction). The inclusion criteria for subjects were limited to studies involving human subjects with stroke. Overall, ten studies were selected to review [4,8,14,15,21-26].
